# FEW-SHOT INTENT INFERENCE
# VIA META-INVERSE REINFORCEMENT LEARNING

## ABSTRACT

A significant challenge for the practical application of reinforcement learning to real world problems is the need to specify an oracle reward function that correctly defines a task. Inverse reinforcement learning (IRL) seeks to avoid this challenge by instead inferring a reward function from expert behavior. While appealing, it can be impractically expensive to collect datasets of demonstrations that cover the variation common in the real world (e.g. opening any type of door). Thus in practice, IRL must commonly be performed with only a limited set of demonstrations where it can be exceedingly difficult to unambiguously recover a reward function. In this work, we exploit the insight that demonstrations from other tasks can be used to constrain the set of possible reward functions by learning a "prior" that is specifically optimized for the ability to infer expressive reward functions from limited numbers of demonstrations. We demonstrate that our method can efficiently recover rewards from images for novel tasks and provide intuition as to how our approach is analogous to learning a prior.

## 1 INTRODUCTION

Reinforcement learning (RL) algorithms have the potential to automate a wide range of decision-making and control tasks across a variety of different domains, as demonstrated by successful recent applications ranging from robotic control (Kober & Peters, 2012; Levine et al., 2016) to game playing (Mnih et al., 2015; Silver et al., 2016). A key assumption of the RL problem statement is the availability of a reward function that accurately describes the desired tasks. For many real world tasks, reward functions can be challenging to manually specify, while being crucial to good performance (Amodei et al., 2016). Most real world tasks are multifaceted and require reasoning over multiple factors in a task (e.g. an autonomous vehicle navigating a city at night), while simultaneously providing appropriate reward shaping to make the task feasible with tractable exploration (Ng et al., 1999). These challenges are compounded by the inherent difficulty of specifying rewards for tasks with high-dimensional observation spaces such as images.

Inverse reinforcement learning (IRL) is an approach that aims to address this problem by instead inferring the reward function from demonstrations of the task (Ng & Russell, 2000). This has the appealing benefit of taking a data-driven approach to reward specification in place of hand engineering. In practice however, rewards functions are rarely learned as it can be prohibitively expensive to provide demonstrations that cover the variability common in real world tasks (e.g., collecting demonstrations of opening every type of door knob). In addition, while learning a complex function from high dimensional observations might make an expressive function approximator seem like a reasonable modelling assumption, in the "few-shot" domain it is notoriously difficult to unambiguously recover a good reward function with expressive function approximators. Prior solutions have thus instead relied on low-dimensional linear models with handcrafted features that effectively encode a strong prior on the relevant features of a task. This requires engineering a set of features by hand that work well for a specific problem. In this work, we propose an approach that instead explicitly learns expressive features that are robust even when learning with limited demonstrations.

Our approach relies on the key observation that related tasks share common structure that we can leverage when learning new tasks. To illustrate, considering a robot navigating through a home. While the exact reward function we provide to the robot may differ depending on the task, there is a structure amid the space of useful behaviours, such as navigating to a series of landmarks, and

there are certain behaviors we *always* want to encourage or discourage, such as avoiding obstacles or staying a reasonable distance from humans. This notion agrees with our understanding of why humans can easily infer the intents and goals (i.e., reward functions) of even abstract agents from just one or a few demonstrations Baker et al. (2007), as humans have access to strong priors about how other humans accomplish similar tasks accrued over many years. Similarly, our objective is to discover the common structure among different tasks, and encode the structure in a way that can be used to infer reward functions from a few demonstrations.

More specifically, in this work we assume access to a set of tasks, along with demonstrations of the desired behaviors for those tasks, which we refer to as the *meta-training set*. From these tasks, we then learn a reward function parameterization that enables effective few-shot learning when used to initialize IRL in a novel task. Our method is summarized in Fig. 1. Our key contribution is an algorithm that enables efficient learning of new reward functions by using meta-training to build a rich "prior" for goal inference. Using our proposed approach, we show that we can learn deep neural network reward functions from raw pixel observations with substantially better data efficiency than existing methods and standard baselines.

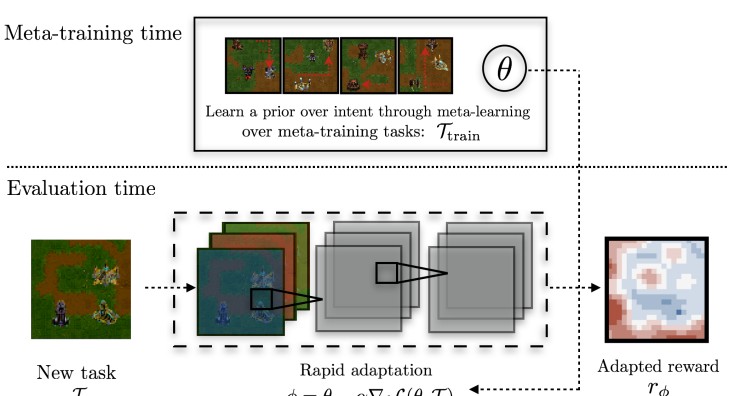

Figure 1: A diagram of our meta-inverse RL approach. Our approach attempts to remedy over-fitting in few-shot IRL by learning a "prior" that constraints the set of possible reward functions to lie within a few steps of gradient descent. Standard IRL attempts to recover the reward function directly from the available demonstrations. The shortcoming of this approach is that there is little reason to expect generalization as it is analogous to training a density model with only a few examples.

## 2 RELATED WORK

Inverse reinforcement learning (IRL) (Ng & Russell, 2000) is the problem of inferring an expert's reward function directly from demonstrations. Prior methods for performing IRL range from margin based approaches (Abbeel & Ng, 2004; Ratliff et al., 2006) to probabilistic approaches (Ramachandran & Amir, 2007; Ziebart et al., 2008). Although it is possible to extend our approach to any other IRL method, in this work we base on work on the maximum entropy (MaxEnt) framework (Ziebart et al., 2008). In addition to allowing for sub-optimality in the expert demonstrations, MaxEnt-IRL can be re-framed as a maximum likelihood estimation problem. 4).

In part to combat the under-specified nature of IRL, prior work has often used low-dimensional linear parameterizations with handcrafted features (Abbeel & Ng, 2004; Ziebart et al., 2008). In order to learn from high dimensional input, Wulfmeier et al. (2015) proposed applying fully convolutional networks (Shelhamer et al., 2017) to the MaxEnt IRL framework (Ziebart et al., 2008) for several navigation tasks (Wulfmeier et al., 2016a;b). Other methods that have incorporated neural network rewards include guided cost learning (GCL) (Finn et al., 2017a), which uses importance sampling and regularization for scalability to high-dimensional spaces, and adversarial IRL (Fu et al., 2018). Several other methods have also proposed imitation learning approaches based on adversarial frameworks that resemble IRL, but do not aim to directly recover a reward function (Ho & Ermon, 2016; Li et al., 2017; Hausman et al., 2017; Kuefler & Kochenderfer, 2018). In this work, instead of improving the ability to learn reward functions on a single task, we focus on the problem of effectively learning to use prior demonstration data from other IRL tasks, allowing us to learn new tasks from a limited number demonstrations even with expressive non-linear reward functions.

Prior work has explored the problem of *multi-task* IRL, where the demonstrated behavior is assumed to have originated from multiple experts achieving different goals. Some of these approaches include those that aim to incorporate a shared prior over reward functions through extending the Bayesian

IRL (Ramachandran & Amir, 2007) framework to the multi-task setting (Dimitrakakis & Rothkopf, 2012; Choi & Kim, 2012). Other approaches have clustered demonstrations while simultaneously inferring reward functions for each cluster (Babeş-Vroman et al., 2011) or introduced regularization between rewards to a common "shared reward" (Li & Burdick, 2017). Our work is similar in that we also seek to encode prior information common to the tasks. However, a critical difference is that our method specifically aims to distill the meta-training tasks into a prior that can then be used to learn rewards for *new* tasks efficiently. The goal therefore is not to acquire good reward functions that explain the meta-training tasks, but rather to use them to learn efficiently on new tasks.

Our approach builds on work on the broader problem of meta-learning (Schmidhuber, 1987; Bengio et al.; Naik & Mammone, 1992; Thrun & Pratt, 2012) and generative modelling (Rezende et al., 2016; Reed et al., 2018; Mordatch, 2018). Prior work has proposed a variety of solutions for learning to learn including memory based methods (Duan et al., 2016; Santoro et al., 2016; Wang et al., 2016; Mishra et al., 2017), methods that learn an optimizer and/or initialization (Andrychowicz et al., 2016; Ravi & Larochelle, 2016; Finn et al., 2017a; Li & Malik, 2017), and methods that compare new datapoints in a learned metric space (Koch, 2015; Vinyals et al., 2016; Shyam et al., 2017; Snell et al., 2017). Our work is motivated by the goal of broadening the applicability of IRL, but in principle it is possible to adapt many of these meta-learning approaches for our problem statement. We leave it to future work to do a comprehensive investigation of different meta-learning approaches which could broaden the applicability of IRL.

## 3 PRELIMINARIES AND OVERVIEW

In this section, we introduce our notation and describe the IRL and meta-learning problems.

### 3.1 LEARNING REWARDS VIA INVERSE REINFORCEMENT LEARNING

The standard Markov decision process (MDP) is defined by the tuple $(\mathcal{S}, \mathcal{A}, p_{\mathbf{s}}, r, \gamma)$ where $\mathcal{S}$ and $\mathcal{A}$ denote the set of possible states and actions respectively, $r : \mathcal{S} \times \mathcal{A} \to \mathbb{R}$ is the reward function, $\gamma \in [0, 1]$ is the discount factor and $p_{\mathbf{s}} : \mathcal{S} \times \mathcal{S} \times \mathcal{A} \to [0, 1]$ denotes the transition distribution over the next state $\mathbf{s}_{t+1}$, given the current state $\mathbf{s}_t$ and current action $\mathbf{a}_t$. Typically, the goal of "forward" RL is to maximize the expected discounted return $R(\tau) = \sum_{t=1}^{T} \gamma^{t-1} r(\mathbf{s}_t, \mathbf{a}_t)$.

In IRL, we instead assume that the reward function is unknown but that we instead have access to a set of expert demonstrations $\mathcal{D} = \{\tau_1, \ldots, \tau_K\}$, where $\tau_k = \{\mathbf{s}_1, \mathbf{a}_1, \ldots, \mathbf{s}_T, \mathbf{a}_T\}$.

The goal of IRL is to recover the unknown reward function $r$ from the set of demonstrations. We build on the maximum entropy (MaxEnt) IRL framework by Ziebart et al. (2008), which models the probability of the trajectories as being distributed proportional to their exponentiated return

$$p(\tau) = \frac{1}{Z} \exp\left(R(\tau)\right), \tag{1}$$

where $Z$ is the partition function, $Z = \int_{\tau} \exp(R(\tau)) \mathrm{d}\tau$. This distribution can be shown to be induced by the optimal policy in entropy regularized forward RL problem:

$$\pi^* =_{\pi} \mathbb{E}_{\tau \sim \pi} \left[R(\tau) - \log \pi(\tau)\right]. \tag{2}$$

This formulation allows us to pose the reward learning problem as a maximum likelihood estimation (MLE) problem in an energy-based model $r_{\phi}$:

$$\min_{\phi} \mathbb{E}_{\tau \sim \mathcal{D}} \left[\mathcal{L}_{\mathrm{IRL}}(\tau)\right] = \min_{\phi} \mathbb{E}_{\tau \sim \mathcal{D}} \left[-\log p_{\phi}(\tau)\right]. \tag{3}$$

Learning in general energy-based models of this form is common in many applications such as structured prediction. However, in contrast to applications where learning can be supervised by millions of labels (e.g. semantic segmentation), the learning problem in Eq. 3 must typically be performed with a relatively small number of example demonstrations. In this work, we seek to address this issue in IRL by providing a way to integrate information from prior tasks to constrain the optimization in Eq. 3 in the regime of limited demonstrations.

### 3.2 META-LEARNING

The goal of meta-learning algorithms is to optimize for the ability to learn efficiently on new tasks. Rather than attempt to generalize to new datapoints, meta-learning can be understood as attempting

to generalize to new tasks. It is assumed in the meta-learning setting that there are two sets of tasks that we refer to as the meta-training set $\{\mathcal{T}_i \; ; \; i = 1..N\}$ and meta-test set $\{\mathcal{T}_j \; ; \; j = 1..M\}$, which are both drawn from a distribution $p(\mathcal{T})$. During meta-training time, the meta-learner attempts to learn the structure of the tasks in the meta-training set, such that when it is presented with a new test task, it can leverage this structure to learn efficiently from a limited number of examples.

To illustrate this distinction, consider the case of few-shot learning setting. Let $f_{\boldsymbol{\theta}}$ denote the learner, and let a task be defined by learning from $K$ training examples $X_{\mathcal{T}}^{\text{tr}} = \{\mathbf{x}_1 \ldots, \mathbf{x}_K\}$, $Y_{\mathcal{T}}^{\text{tr}} = \{\mathbf{y}_1 \ldots, \mathbf{y}_K\}$, and evaluating on $K'$ test examples $X_{\mathcal{T}}^{\text{test}} = \{\mathbf{x}_1 \ldots, \mathbf{x}_{K'}\}$, $Y_{\mathcal{T}}^{\text{test}} = \{\mathbf{y}_1 \ldots, \mathbf{y}_{K'}\}$. One approach to meta-learning is to directly parameterize the meta-learner with an expressive model such as a recurrent or recursive neural network (Duan et al., 2016; Mishra et al., 2017) conditioned on the task training data and the inputs for the test task: $f_{\boldsymbol{\theta}}(Y | X_{\mathcal{T}}^{\text{test}}, X_{\mathcal{T}}^{\text{tr}}, Y_{\mathcal{T}}^{\text{tr}})$. Such a model is optimized using log-likelihood across all tasks. In this approach to meta-learning, since neural networks are known to be universal function approximators (Siegelmann & Sontag, 1995), any desired structure between tasks can be implicitly encoded.

Rather than learn a single black-box function, another approach to meta-learning is to learn components of the learning procedure such as the initialization (Finn et al., 2017a) or the optimization algorithm (Ravi & Larochelle, 2016; Andrychowicz et al., 2016). In this work we extend the approach of model agnostic meta-learning (MAML) introduced by Finn et al. (2017a), which learns an initialization that is adapted by gradient descent. Concretely, in the supervised learning case, given a loss function $\mathcal{L}(\boldsymbol{\theta}, X_{\mathcal{T}}, Y_{\mathcal{T}})$ (e.g. cross-entropy), MAML performs the following optimization

$$\min_{\boldsymbol{\theta}} \sum_{\mathcal{T}} \mathcal{L}(\phi_{\mathcal{T}}, X_{\mathcal{T}}^{\text{test}}, Y_{\mathcal{T}}^{\text{test}}) = \min_{\boldsymbol{\theta}} \sum_{\mathcal{T}} \mathcal{L} \left( \boldsymbol{\theta} - \alpha \nabla_{\boldsymbol{\theta}} \mathcal{L}(\boldsymbol{\theta}, X_{\mathcal{T}}^{\text{tr}}, Y_{\mathcal{T}}^{\text{tr}}), X_{\mathcal{T}}^{\text{test}}, Y_{\mathcal{T}}^{\text{test}} \right), \qquad (4)$$

where the optimization is over an initial set of parameters $\boldsymbol{\theta}$ and the loss on the held out tasks $X_{\mathcal{T}}^{test}$ becomes the signal for learning the initial parameters for gradient descent on $X_{\mathcal{T}}^{tr}$. This optimization is analogous to adding a constraint in a multi-task setting, which we show in later sections is analogous in our setting to learning a prior over reward functions.

# 4 LEARNING TO LEARN REWARDS

Our goal in meta-IRL is to learn how to learn reward functions across many tasks such that the model can infer the reward function for a new task using only one or a few expert demonstrations. Intuitively, we can view this problem as aiming to learn a prior over the intentions of human demonstrators, such that when given just one or a few demonstrations of a new task, we can combine the learned prior with the new data to effectively determine the human's reward function. Such a prior is helpful in inverse reinforcement learning settings, since the space of relevant reward functions is much smaller than the space of all possible rewards definable on the raw observations.

During meta-training, we have a set of tasks $\{\mathcal{T}_i \; ; \; i = 1..N\}$. Each task $\mathcal{T}_i$ has a set of demonstrations $\mathcal{D}_{\mathcal{T}} = \{\tau_1, \ldots, \tau_K\}$ from an expert policy which we partition into disjoint $\mathcal{D}_{\mathcal{T}}^{\text{tr}}$ and $\mathcal{D}_{\mathcal{T}}^{\text{test}}$ sets. The demonstrations for each meta-training task are assumed to be produced by the expert according to the maximum entropy model in Section 3.1. During meta-training, these tasks will be used to encodes common structure so that our model can quickly acquire rewards for new tasks from just a few demonstrations.

After meta-training, our method is presented with a new task. During this meta-test phase, the algorithm must infer the parameters of the reward function $r_{\phi}(\mathbf{s}_t, \mathbf{a}_t)$ for the new task from a few demonstrations. As is standard in meta-learning, we assume that the test task is from the same distribution of tasks seen during meta-training, a distribution that we denote as $p(\mathcal{T})$.

## 4.1 META REWARD AND INTENTION LEARNING (MANDRIL)

In order to meta-learn a reward function that can act as a prior for new tasks and new environments, we first formalize the notion of a good reward by defining a loss $\mathcal{L}_{\mathcal{T}}(\boldsymbol{\theta})$ on the reward function $r_{\boldsymbol{\theta}}$ for a particular task $\mathcal{T}$. We use the MaxEnt IRL loss $\mathcal{L}_{\text{IRL}}$ discussed in Section 3, which, for a given $\mathcal{D}_{\mathcal{T}}$, leads to the following gradient (Ziebart et al., 2008):

$$\nabla_{\boldsymbol{\theta}} \mathcal{L}_{\mathcal{T}}(\boldsymbol{\theta}) = \frac{\partial r_{\boldsymbol{\theta}}}{\partial \boldsymbol{\theta}} \left[ \mathbb{E}_{\tau}[\boldsymbol{\mu}_{\tau}] - \boldsymbol{\mu}_{\mathcal{D}_{\mathcal{T}}} \right]. \qquad (5)$$

---

**Algorithm 1** Meta Reward and Intention Learning (MandRIL)

---

1: **Input:** Set of meta-training tasks $\{\mathcal{T}\}^{\text{meta-train}}$
2: **Input:** hyperparameters $\alpha, \beta$
3: **function** MAXENTIRL-GRAD($r_{\boldsymbol{\theta}}, \mathcal{T}, \mathcal{D}$)          ▷ Single task update
4:    $\boldsymbol{\mu}_{\mathcal{D}} = $ STATE-VISITATIONS-TRAJ($\mathcal{T}, \mathcal{D}$)     ▷ Compute state visitations of demos
5:    $\mathbb{E}_{\tau}[\boldsymbol{\mu}_{\tau}] = $ STATE-VISITATIONS-POLICY($r_{\boldsymbol{\theta}}, \mathcal{T}$)    ▷ Compute Max-Ent state visitations
6:    $\frac{\partial \mathcal{L}}{\partial r_{\boldsymbol{\theta}}} = \mathbb{E}_{\tau}[\boldsymbol{\mu}_{\tau}] - \boldsymbol{\mu}_{\mathcal{D}}$      ▷ MaxEntIRL gradient (Ziebart et al., 2008)
7:    **Return** $\frac{\partial \mathcal{L}}{\partial r_{\boldsymbol{\theta}}}$

8:
9: Randomly initialize $\boldsymbol{\theta}$
10: **while** not done **do**
11:    Sample batch of tasks $\mathcal{T}_i \sim \{\mathcal{T}\}^{\text{meta-train}}$
12:    **for all** $\mathcal{T}_i$ **do**
13:      Sample demos $\mathcal{D}^{\text{tr}} = \{\tau_1, \ldots, \tau_K\} \sim \mathcal{T}_i$
14:      $\frac{\partial \mathcal{L}_{\mathcal{T}_i}^{\text{tr}}(\boldsymbol{\theta})}{\partial r_{\boldsymbol{\theta}}} = $ MAXENTIRL-GRAD($r_{\boldsymbol{\theta}}, \mathcal{T}_i, \mathcal{D}^{\text{tr}}$)    ▷ Inner loss computation
15:      Compute $\nabla_{\boldsymbol{\theta}} \mathcal{L}_{\mathcal{T}_i}^{\text{tr}}(\boldsymbol{\theta})$ from $\frac{\partial \mathcal{L}_{\mathcal{T}_i}^{\text{tr}}(\boldsymbol{\theta})}{\partial r_{\boldsymbol{\theta}}}$ via chain rule
16:      Compute updated parameters $\boldsymbol{\phi}_{\mathcal{T}_i} = \boldsymbol{\theta} - \alpha \nabla_{\boldsymbol{\theta}} \mathcal{L}_{\mathcal{T}_i}^{\text{tr}}(\boldsymbol{\theta})$     ▷ Fast update
17:      Sample demos $\mathcal{D}^{\text{test}} = \{\tau_1', \ldots, \tau_{K'}'\} \sim \mathcal{T}_i$
18:      $\frac{\partial \mathcal{L}_{\mathcal{T}_i}^{\text{test}}}{\partial r_{\boldsymbol{\theta}}} = $ MAXENTIRL-GRAD($r_{\boldsymbol{\phi}_{\mathcal{T}_i}}, \mathcal{T}_i, \mathcal{D}^{\text{test}}$))    ▷ Outer loss computation
19:      Compute $\nabla_{\boldsymbol{\theta}} \mathcal{L}_{\mathcal{T}_i}^{\text{test}}$ from $\frac{\partial \mathcal{L}_{\mathcal{T}_i}^{\text{test}}}{\partial r_{\boldsymbol{\theta}}}$ via chain rule    ▷ Compute meta-gradient
20:    Compute update to $\boldsymbol{\theta} \leftarrow \boldsymbol{\theta} - \beta \sum_i \nabla_{\boldsymbol{\theta}} \mathcal{L}_{\mathcal{T}_i}^{\text{test}}$    ▷ Update initial parameters

---

where $\boldsymbol{\mu}_{\tau}$ are the state visitations under the optimal maximum entropy policy under $r_{\boldsymbol{\theta}}$, and $\boldsymbol{\mu}_{\mathcal{D}}$ are the mean state visitations under the demonstrated trajectories.

If our end goal were to achieve a single reward function that works as well as possible across all tasks in $\{\mathcal{T}_i \; ; \; i = 1..N\}$, then we could simply follow the *mean* gradient across all tasks. However, our objective is different: instead of optimizing performance on the meta-training tasks, we aim to learn a reward function that can be quickly and efficiently adapted to new tasks at meta-test time. In doing so, we aim to encode prior information over the task distribution in this learned reward prior.

We propose to implement such a learning algorithm by finding the parameters $\boldsymbol{\theta}$, such that starting from $\boldsymbol{\theta}$ and taking a small number of gradient steps on a few demonstrations from given task leads to a reward function for which a set of *test* demonstrations have high likelihood, with respect to the MaxEnt IRL model. In particular, we would like to find a $\boldsymbol{\theta}$ such that the parameters

$$\boldsymbol{\phi}_{\mathcal{T}} = \boldsymbol{\theta} - \alpha \nabla_{\boldsymbol{\theta}} \mathcal{L}_{\mathcal{T}}^{\text{tr}}(\boldsymbol{\theta}) \tag{6}$$

lead to a reward function $r_{\boldsymbol{\phi}_{\mathcal{T}}}$ for task $\mathcal{T}$, such that the IRL loss (corresponding to negative log-likelihood) for a disjoint set of test demonstrations, given by $\mathcal{L}_{\text{IRL}}^{\mathcal{T}, \text{test}}$, is minimized. The corresponding optimization problem for $\boldsymbol{\theta}$ can therefore be written as follows:

$$\min_{\boldsymbol{\theta}} \sum_{i=1}^{N} \mathcal{L}_{\mathcal{T}_i}^{\text{test}}(\boldsymbol{\phi}_{\mathcal{T}_i}) = \sum_{i=1}^{N} \mathcal{L}_{\mathcal{T}_i}^{\text{test}} \left( \boldsymbol{\theta} - \alpha \nabla_{\boldsymbol{\theta}} \mathcal{L}_{\mathcal{T}_i}^{\text{tr}}(\boldsymbol{\theta}) \right). \tag{7}$$

Our method acquires this prior $\boldsymbol{\theta}$ over rewards in the task distribution $p(\mathcal{T})$ by optimizing this loss. This amounts to an extension of the MAML algorithm in Section 3.2 to the inverse reinforcement learning setting. This extension is quite challenging, because computing the MaxEnt IRL gradient requires repeatedly solving for the current maximum entropy policy and visitation frequencies, and the MAML objective requires computing derivatives *through* this gradient step. Next, we describe in detail how this is done. An overview of our method is also outlined in Alg. 1.

**Meta-training.** The computation of the meta-gradient for the objective in Eq. 7 can be conceptually separated into two parts. First, we perform the update in Eq. 6 by computing the *expected state visitations* $\boldsymbol{\mu}$, which is the expected number of times an agent will visit each state. We denote this overall procedure as STATE-VISITATIONS-POLICY, and follow Ziebart et al. (2008) by first computing the maximum entropy optimal policy in Eq. 2 under the current $r_{\boldsymbol{\theta}}$, and then approximating $\boldsymbol{\mu}$

using dynamic programming. Next, we compute the state visitation distribution of the expert using a procedure which we denote as STATE-VISITATIONS-TRAJ. This can be done either empirically, by averaging the state visitation of the experts demonstrations, or by using STATE-VISITATIONS-POLICY if the true reward is available at meta-training time. This allows us to recover the IRL gradient according to Eq. 5, which we can then apply to compute $\phi_{\mathcal{T}}$ according to Eq. 6.

Second, we need to differentiate through this update to compute the gradient of the meta-loss in Eq. 7. Note that the meta-loss itself is the IRL loss evaluated with a different set of test demonstrations. We follow the same procedure as above to evaluate the gradient of $\mathcal{L}_{\mathrm{IRL}}^{\mathcal{T},\mathrm{test}}$ with respect to the post-update parameters $\phi_{\mathcal{T}}$, and then apply the chain rule to compute the meta-gradient:

$$
\begin{aligned}
\nabla_{\boldsymbol{\theta}} \mathcal{L}_{\mathcal{T}}^{\mathrm{test}}(\boldsymbol{\theta}) &= \frac{\partial \mathcal{L}_{\mathcal{T}}^{\mathrm{test}}}{\partial r_{\phi_{\mathcal{T}}}} \frac{\partial r_{\phi_{\mathcal{T}}}}{\partial \phi_{\mathcal{T}}} \frac{\partial}{\partial \boldsymbol{\theta}} (\boldsymbol{\theta} - \alpha \nabla_{\boldsymbol{\theta}} \mathcal{L}_{\mathcal{T}}^{\mathrm{tr}}(\boldsymbol{\theta})) \\
&= \frac{\partial \mathcal{L}_{\mathcal{T}}^{\mathrm{test}}}{\partial r_{\phi_{\mathcal{T}}}} \frac{\partial r_{\phi_{\mathcal{T}}}}{\partial \phi_{\mathcal{T}}} \left( \mathbf{I} - \alpha \frac{\partial^2 \mathcal{L}_{\mathcal{T}}^{\mathrm{tr}}(\boldsymbol{\theta})}{\partial \boldsymbol{\theta}^2} - \alpha \frac{\partial r_{\boldsymbol{\theta}}}{\partial \boldsymbol{\theta}} \frac{\partial r_{\boldsymbol{\theta}}}{\partial \boldsymbol{\theta}}^{\top} \frac{\partial}{\partial r_{\boldsymbol{\theta}}} \mathbb{E}_{\tau}[\boldsymbol{\mu}_{\tau}] \right)
\end{aligned}
\tag{8}
$$

where on the second line we differentiate through the MaxEnt-IRL update. The derivation of this expression is somewhat more involved and provided in the supplementary Appendix D.

**Meta-testing.** Once we have acquired the meta-trained parameters $\boldsymbol{\theta}$ that encode a prior over $p(\mathcal{T})$, we can leverage this prior to enable fast, few-shot IRL of novel tasks in $\{\mathcal{T}_j \; ; \; j = 1..M\}$. For each task, we first compute the state visitations from the available set of demonstrations for that task. Next, we use these state visitations to compute the gradient, which is the same as the inner loss gradient computation of the meta-training loop in Alg. 1. We apply this gradient to adapt the parameters $\boldsymbol{\theta}$ to the new task. Even if the model was trained with only one to three inner gradient steps, we found in practice that it was beneficial to take substantially more gradient steps during meta-testing; performance continued to improve with up to 20 steps.

## 4.2 INTERPRETATION AS LEARNING A PRIOR OVER INTENT

The objective in Eq. 6 optimizes for parameters that enable that reward function to adapt and generalize efficiently on a wide range of tasks. Intuitively, constraining the space of reward functions to lie within a few steps of gradient descent can be interpreted as expressing a "locality" prior over reward function parameters. This intuition can be made more concrete with the following analysis.

By viewing IRL as maximum likelihood estimation, we can take the perspective of Grant et al. (2018) who showed that for a linear model, fast adaptation via a few steps of gradient descent in MAML is performing MAP inference over $\phi$, under a Gaussian prior with the mean $\boldsymbol{\theta}$ and a covariance that depends on the step size, number of steps and curvature of the loss. This is based on the connection between early stopping and regularization previously discussed in Santos (1996), which we refer the readers to for a more detailed discussion. The interpretation of MAML as imposing a Gaussian prior on the parameters is exact in the case of a likelihood that is quadratic in the parameters (such as the log-likelihood of a Gaussian in terms of its mean). For any non-quadratic likelihood, this is an approximation in a local neighborhood around $\boldsymbol{\theta}$ (i.e. up to convex quadratic approximation). In the case of very complex parameterizations, such

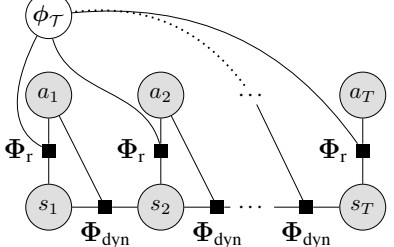

Figure 2: Our approach can be understood as approximately learning a distribution over the demonstrations $\tau$, in the factor graph $p(\tau) = \frac{1}{Z} \prod_{t=1}^{T} \boldsymbol{\Phi}_{\mathrm{r}}(\phi_{\mathcal{T}}, \mathbf{s}_t, \mathbf{a}_t) \boldsymbol{\Phi}_{\mathrm{dyn}}(\mathbf{s}_{t+1}, \mathbf{s}_t, \mathbf{a}_t)$ (above) where we learn a prior over $\phi_{\mathcal{T}}$, which during meta-test is used for MAP inference over new expert demonstrations.

as deep function approximators, this is a coarse approximation and unlikely to be the mode of a posterior. However, we can still frame the effect of early stopping and initialization as serving as a prior in a similar way as prior work (Sjöberg & Ljung, 1995; Duvenaud et al., 2016; Grant et al., 2018). More importantly, this interpretation hints at future extensions to our approach that could benefit from employing more fully Bayesian approaches to reward and goal inference.

## 5 EXPERIMENTS

Our evaluation seeks to answer two questions. First, we aim to test our core hypothesis that leveraging prior task experience enables reward learning for new tasks with just a few demonstrations. Second, we compare our method with alternative algorithms that make use of multi-task experience.

We test our core hypothesis by comparing learning performance on a new task starting from the learned initialization produced by MandRIL, compared to starting from scratch with a random initialization. This comparison is meant to evaluate whether prior experience on other tasks can in fact make inverse RL more efficient.

To our knowledge, there is no prior work that addresses the meta-inverse reinforcement learning problem introduced in this paper. Thus, to provide a point of comparison and calibrate the difficulty of the tasks, we adapt two alternative black-box meta-learning methods to the IRL setting. The comparisons to both of the black-box methods described below evaluate the importance of incorporating the IRL gradient into the meta-learning process, rather than learning the adaptation process entirely from scratch.

- **Demo conditional model**: Our method implicitly conditions on the demonstrations through the gradient update. In principle, a conditional deep model with sufficient capacity could implicitly implement a similar learning rule. Thus, we consider a conditional model (often referred to as a "contextual model" (Finn et al., 2017b)), which receives the demonstration as an additional input.

- **Recurrent meta-learner**: We additionally compare to an RNN-based meta-learner (Santoro et al., 2016; Duan et al., 2017). Specifically, we implement a conditional model by feeding both images and sequences of states visited by the demonstrations to an LSTM.

Our approach can be understood as explicitly optimizing for an effective parameter initialization for the IRL problem. In order to test the benefits of our proposed formulation, we also compare with finetuning an initialization obtained with the same set of prior tasks, but with supervised pretraining as follows:

- **Supervised pre-training:** We compare to following the average gradient during meta-training, averaged across tasks, and fine-tuning at meta-test time (as discussed in Section 4). This comparison evaluates the benefits of optimizing explicitly for weights that perform well under fine-tuning. We compare to pre-training on a single task as well as all the meta-training tasks.

Next, we describe our environment and evaluation.

**Spriteworld navigation domain.** Since most prior IRL works (and multi-task IRL works) have studied settings where linear reward function approximators suffice (i.e. low-dimensional state spaces made up from hand-designed features), we design an experiment that is significantly more challenging—that requires learning rewards on raw pixels—while still exhibiting multi-task structure needed to test our core hypothesis. We consider a navigation problem where we aim to learn a convolutional neural network that directly maps image pixels to rewards. To do so, we introduce "SpriteWorld," which is a synthetically generated task, some examples of which are shown in Fig. 3. The task visuals are inspired by Starcraft and work applying learning algorithms to perform micromanagement (e.g. (Synnaeve et al., 2016)), although we do not use the game engine. Tasks involve navi-

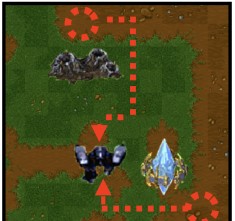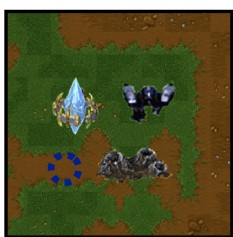

Figure 3: An example task: When learning a task, the agent has access to the image (left) and demonstrations (red arrows). To evaluate the agent's learning (right), the agent is tested for its ability to recover the reward for the task when the objects have been rearranged. The reward structure we wish to capture can be illustrated by considering the initial state in blue. An policy acting optimally under a correctly inferred reward should interpret the other objects as obstacles, and prefer a path on the dirt in between them.

gating to goal objects while exhibiting preference over terrain types (e.g. the agent prefers to traverse dirt tiles over traversing grass tiles). At meta-test time, we provide one or a few demonstrations in a single training environment and evaluate the reward learned using these demonstrations in a new, test environment that contains the same objects as the training environment, but arranged differently. Evaluating in a new test environment is critical to measure that the reward learned the correct visual cues, rather than simply memorizing the demonstration trajectory.

The underlying MDP structure of SpriteWorld is a grid, where the states are each of the grid cells, and the actions enable the agent to move to any one of its 8-connected neighbors. We generate unique tasks from this domain as follows. First, we randomly choose a set of 3 sprites from a total of 100 sprites from the original game (creating a total of 161,700 unique tasks). We randomly place these three sprites within a randomly generated terrain tiling; we designate one of the sprites to be the goal of the navigation task, i.e. the object to which the agent must navigate. The other two objects are treated as obstacles for which the agent incurs a large negative reward for not avoiding. In each task, we optimize our model on a meta-training set and evaluate the ability of the reward function to the generalize to a rearrangement of the same objects. For example, suppose the goal of the task is to navigate to sprite A, while avoiding sprites B and C. Then, to generate an environment for evaluation, we resample the positions of the sprites, while the underlying task remains the same (i.e., navigate to A). This requires the model to make use of the visual patterns in the scene to generalize effectively, rather than simply memorizing positions of sprites. We evaluate on novel combinations of units seen in meta-training, as well as the ability to generalize to new unseen units. We provide further details on our setup in Appendices A and B.

We measure performance using the expected value difference, which measures the sub-optimality of a policy learned under the learned reward; this is a standard performance metric used in prior IRL work (Levine et al., 2011; Wulfmeier et al., 2015). The metric is computed by taking the difference between the value of the optimal policy under the learned reward and the value of the optimal policy under the true reward.

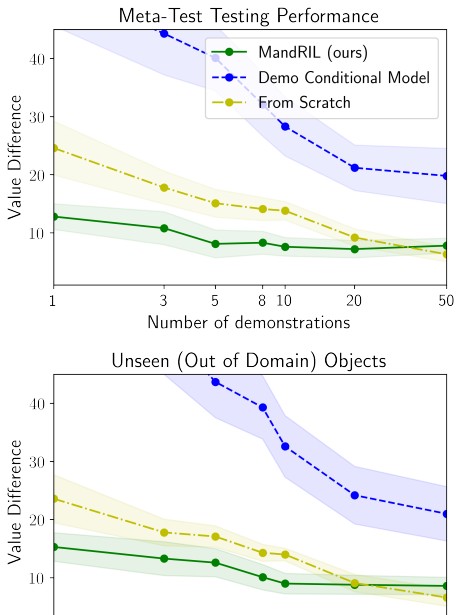

Figure 4: Meta-test performance with varying numbers of demonstrations (lower is better): held-out tasks test performance (top), and test performance on held-out tasks with novel sprites (bottom). All methods are capable are overfitting to the training environment (See Appendix A). However, in both test settings, MandRIL achieves comparable performance to the training environment, while the other methods overfit until they receive at least 10 demonstrations. The recurrent meta-learner has a value difference larger than 60 in both test settings. Shaded regions show 95% confidence intervals.

**Evaluation protocol.** We evaluate on held-out tasks that were unseen during meta-training. We consider two settings: (1) tasks involving new combinations and placements of sprites, with individual sprites that were present during meta-training, and (2) tasks with combinations of entirely new sprites which we refer to as "out of domain objects." For each task, we generate one environment (set of sprite positions) along with one or a few demonstrations for adapting the reward, and generate a second environment (with new sprite positions) where we evaluate the adapted reward. In these meta-test time evaluations, we refer to the performance on the first environment as "training performance" (not to be confused with meta-training) and to performance on the second as "testing performance". We evaluate on 32 meta-test randomly generated tasks.

**Results.** The results are shown in Fig. 4, which illustrate test performance with in-distribution and out-of-distribution sprites. Our approach, MandRIL, achieves consistently better performance in both settings. Most significantly, our approach performs well even with single-digit numbers of demonstrations. By comparison, alternative meta-learning methods generally overfit considerably, attaining good training performance (see Appendix. A for curves) but poor test performance. Learning the reward function from scratch is in fact the most competitive baseline – as the number of demonstrations increases, simply training the fully convolutional reward function from scratch on the new task is the only method that matches the performance of MandRIL when provided 20 or more demonstrations. However, with only a few demonstrations, MandRIL has substantially lower value difference. It is worth noting the performance of MandRIL on the out of distribution test setting (Fig. 4, bottom): although the evaluation is on new sprites, MandRIL is still able to adapt via gradient descent and exceed the performance of learning from scratch and all other methods.

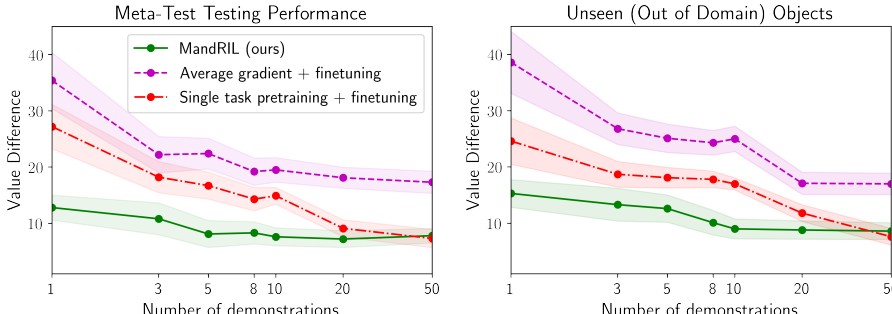

Figure 5: Meta-test performance comparison with varying number of demonstrations using pre-trained weights fine-tuned over either a single task or all tasks. We find that pre-training on the full set of tasks hurts adaptation, while pre-training on a single task does not improve performance (comparable to random initialization). Our approach ManDRIL outperforms all these methods, which shows that explicitly optimizing for initial weights for fine-tuning robustly improves performance. Shaded regions show 95% confidence intervals.

Finally, as it is common practice to fine-tune representations obtained from a supervised pre-training phase, we perform this comparison in Figure 5. We compare against an approach that follows the mean gradient across the tasks at meta-training time and is fine-tuned at meta-test time which we find is less effective than learning from a random initialization. We conclude that fine tuning reward functions learned in this manner is not an effective way of using prior task information. When using a single task for pre-training, we find that it performs comparable to random initialization. In contrast, we find that our approach, which explicitly optimizes for initial weights for fine-tuning, robustly improves performance.

## 6 CONCLUSION

In this work, we present an approach that enables few-shot learning for reward functions of new tasks. We achieve this through a novel formulation of inverse reinforcement learning that learns to encode common structure across tasks. Using our meta-IRL approach, we show that we can leverage data from previous tasks to effectively learn deep neural network reward functions from raw pixel observations for new tasks, from only a handful of demonstrations. Our work paves the way for futures work that considers environments with unknown dynamics, or more fully probabilistic approaches to reward and goal inference.

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

# Appendix

## A EXPERIMENTAL DETAILS

The input to our reward function for all experiments is a resized $80 \times 80$ RGB image, with an output space of $20 \times 20$ in the underlying MDP state space $\mathbf{s}$. In order to compute the optimal policy, we use Q-iteration. In our experiments, we parameterize the reward function for all our reward functions starting from the same base learner. The first layer is a $8 \times 8$ convolution with a stride of 2, 256 filters and symmetric padding of 4. The second layer is a $4 \times 4$ with a stride of 2, 128 filters and symmetric of 1. The third and fourth layer are $3 \times 3$ convolutions with a stride of 1, 64 filters and symmetric padding of 1. The final layer is a $1 \times 1$ convolution.

Our LSTM (Hochreiter & Schmidhuber, 1997) implementation is based on the variant used by Zaremba et al. (Zaremba et al., 2014). The input to the LSTM at each time step is the location of the agent in the, we separately embed the $(x, y)$-coordinates. This is then used to predict an additional channel in to the base CNN architecture described above. We also experimented with conditioning the initial hidden state on image features from a separate CNN, but found that this did not improve performance.

In our demo conditional model, we preserve the spatial information of the demonstrations by feeding in the state visitation map as a image-grid, upsampled with bi-linear interpolation, as an additional channel to the image. In our setup, both the demo-conditional models share the same convolutional architecture, but differ only in how they encode condition on the demonstrations.

For all our methods, we optimzed our model with Adam (Kingma & Ba, 2014). We turned over the learning rate $\alpha$, the inner learning rate $\beta$ of our approach and $\ell_2$ weight decay on the initial parameters. In our LSTM learner, we experimented with different embedding sizes, as well as the dimensionality of the LSTM although we found that these hyperparameters did not impact performance. A negative result we found was that bias transformation (Finn et al., 2017b) did not help in our experimental setting.

Table 1: Summary of hyperparameters on Spriteworld environment.

| Hyperparameters | Value |
|---|---|
| Architecture | Conv(256-8x8-2) -Conv(128-4x4-2) -Conv(64-3x3-1) -Conv(64-3x3-1) -Conv(1-1x1-1) |
| Learning rate $\alpha$ | Best chosen from $\{0.0001, 0.00001\}$ |
| Inner learning rate $\beta$ | Best chosen from $\{0.001, 0.0005\}$ |
| Weight decay $\ell_2$ | Best chosen from $\{0, 0.0001\}$ |
| LSTM hidden dimension | Best chosen from $\{128, 256\}$ |
| LSTM embedding sizes | Best chosen from $\{64, 128\}$ |
| Batch size | 16 |
| Number of meta-training environments | 1000 |
| Number of meta-val/test environments | 32 |
| Maximum horizon (T) | 15 |

## B ENVIRONMENT DETAILS

The sprites in our environment are extracted directly from the StarCraft files. We used in total 100 random units for meta-training. Evaluation on new objects was performed with 5 randomly selected sprites. For computational efficiency, we create a meta-training set of 1000 tasks and cache the optimal policy and state visitations under the true cost. Our evaluation is over 32 tasks. Our set of sprites was divided into two categories: buildings and characters. Each characters had multiple poses (taken from different frames of animation, such as walking/running/flying), whereas buildings only had a single pose. During meta-training the units were randomly placed, but to avoid the possibility

that the agent would not need to actively avoid obstacles, the units were placed away from the boundary of the image in both the meta-validation and meta-test set.

The terrain in each environment was randomly generated using a set of tiles, each belonging to a specific category (e.g. grass, dirt, water). For each tile, we also specified a set of possible tiles for each of the 4-neighbors. Using these constraints on the neighbors, we generated random environment terrains using a graph traversal algorithm, where successor tiles were sampled randomly from this set of possible tiles. This process resulted in randomly generated, seamless environments. The names of the units used in our experiments are as follows (names are from the original game files):

The list of buildings used is: academy, assim, barrack, beacon, cerebrat, chemlab, chrysal, cocoon, comsat, control, depot, drydock, egg, extract, factory, fcolony, forge, gateway, genelab, geyser, hatchery, hive, infest, lair, larva, mutapit, nest, nexus, nukesilo, nydustpit, overlord, physics, probe, pylon, prism, pillbox, queen, rcluster, refinery, research, robotic, sbattery, scolony, spire, starbase, stargate, starport, temple, warm, weaponpl, wessel.

The list of characters used is: acritter, arbiter, archives, archon, avenger, battlecr, brood, bugguy, carrier, civilian, defiler, dragoon, drone, dropship, firebat, gencore, ghost, guardian, hydra, intercep, jcritter, lurker, marine, missile, mutacham, mutalid, sapper, scout, scv, shuttle, snakey, spider, stank, tank, templar, trilob, ucereb, uikerr, ultra, vulture, witness, zealot, zergling.

## C  META-TEST TRAINING PERFORMANCE

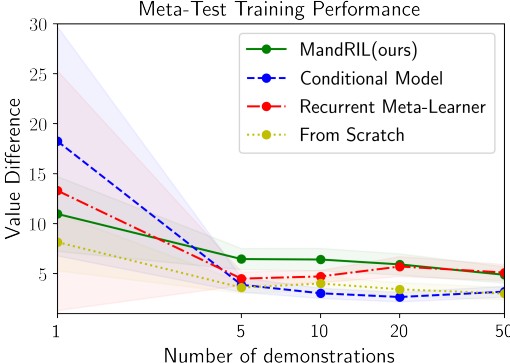

Figure 6: Meta-test "training" performance with varying numbers of demonstrations (lower is better). This is the performance on the environment for which demonstrations are provided for adaptation. As the number of demonstrations increase, all methods are able to perform well in terms of training performance as they can simply overfit to the training environment without acquiring the right visual cues that allow them to generalize. However, we find comes at the cost comes of considerable overfitting as we discuss in Section. 5.

## D  DETAILED META-OBJECTIVE DERIVATION

We define the quality of reward function $r_{\boldsymbol{\theta}}$ parameterized by $\boldsymbol{\theta} \in \mathbb{R}^k$ on task $\mathcal{T}$ with the MaxEnt IRL loss, $\mathcal{L}_{\text{IRL}}^{\mathcal{T}}(\boldsymbol{\theta})$, described in Section 4. The corresponding gradient is

$$\nabla_{\boldsymbol{\theta}} \mathcal{L}_{\text{IRL}}(\boldsymbol{\theta}) = \frac{\partial r_{\boldsymbol{\theta}}}{\partial \boldsymbol{\theta}}(\mathbb{E}_{\tau}[\boldsymbol{\mu}_{\tau}] - \boldsymbol{\mu}_{\mathcal{D}_{\mathcal{T}}}), \tag{9}$$

where $\partial r_{\boldsymbol{\theta}} / \partial \boldsymbol{\theta}$ is the $k \times |\mathcal{S}||\mathcal{A}|$-dimensional Jacobian matrix of the reward function $r_{\boldsymbol{\theta}}$ with respect to the parameters $\boldsymbol{\theta}$. Here, $\boldsymbol{\mu}_{\tau} \in \mathbb{R}^{|\mathcal{S}||\mathcal{A}|}$ is the vector of *state visitations* under the trajectory $\tau$ (i.e. the vector whose elements are 1 if the corresponding state-action pair has been visited by the trajectory $\tau$, and 0 otherwise), and $\boldsymbol{\mu}_{\mathcal{D}_{\mathcal{T}}} = \frac{1}{|\mathcal{D}_{\mathcal{T}}|} \sum_{\tau \in \mathcal{D}_{\mathcal{T}}} \boldsymbol{\mu}_{\tau}$ is the mean state visitations over all demonstrated trajectories in $\mathcal{D}_{\mathcal{T}}$. Let $\boldsymbol{\phi}_{\mathcal{T}} \in \mathbb{R}^k$ be the updated parameters after a single gradient step. Then

$$\boldsymbol{\phi}_{\mathcal{T}} = \boldsymbol{\theta} - \alpha \nabla_{\boldsymbol{\theta}} \mathcal{L}_{\mathcal{T}}^{\text{tr}}(\boldsymbol{\theta}). \tag{10}$$

Let $\mathcal{L}_{\mathcal{T}}^{\text{test}}$ be the MaxEnt IRL loss, where the expectation over trajectories is computed with respect to a test set that is *disjoint* from the set of demonstrations used to compute $\mathcal{L}_{\mathcal{T}}^{\text{test}}(\boldsymbol{\theta})$ in Eq. 10. We seek to minimize

$$\sum_{\mathcal{T} \in \mathcal{T}^{\text{tr}}} \mathcal{L}_{\mathcal{T}}^{\text{test}}(\boldsymbol{\phi}_{\mathcal{T}}) \tag{11}$$

over the parameters $\boldsymbol{\theta}$. To do so, we first compute the gradient of Eq. 11, which we derive here. Applying the chain rule

$$
\begin{aligned}
\nabla_{\boldsymbol{\theta}} \mathcal{L}_{\mathcal{T}}^{\text{test}} &= \frac{\partial \boldsymbol{\phi}_{\mathcal{T}}}{\partial \boldsymbol{\theta}} \frac{\partial r_{\boldsymbol{\phi}_{\mathcal{T}}}}{\partial \boldsymbol{\phi}_{\mathcal{T}}} \frac{\partial \mathcal{L}_{\mathcal{T}}^{\text{test}}}{\partial r_{\boldsymbol{\phi}_{\mathcal{T}}}} \\
&= \frac{\partial}{\partial \boldsymbol{\theta}} \left( \boldsymbol{\theta} - \alpha \nabla_{\boldsymbol{\theta}} \mathcal{L}_{\mathcal{T}}^{\text{tr}}(\boldsymbol{\theta}) \right) \frac{\partial r_{\boldsymbol{\phi}_{\mathcal{T}}}}{\partial \boldsymbol{\phi}_{\mathcal{T}}} \frac{\partial \mathcal{L}_{\mathcal{T}}^{\text{test}}}{\partial r_{\boldsymbol{\phi}_{\mathcal{T}}}} \\
&= \left( \mathbf{I} - \alpha \frac{\partial}{\partial \boldsymbol{\theta}} \left( \frac{\partial r_{\boldsymbol{\theta}}}{\partial \boldsymbol{\theta}} (\mathbb{E}_{\tau}[\boldsymbol{\mu}_{\tau}] - \boldsymbol{\mu}_{\mathcal{D}_{\mathcal{T}}}) \right) \right) \frac{\partial r_{\boldsymbol{\phi}_{\mathcal{T}}}}{\partial \boldsymbol{\phi}_{\mathcal{T}}} \frac{\partial \mathcal{L}_{\mathcal{T}}^{\text{test}}}{\partial r_{\boldsymbol{\phi}_{\mathcal{T}}}}
\end{aligned}
\tag{12}
$$

where in the last line we substitute in the gradient of the MaxEnt IRL loss in Eq. 9. In Eq. 12, we use the following notation:

- $\partial \boldsymbol{\phi}_{\mathcal{T}} / \partial \boldsymbol{\theta}$ denotes the $k \times k$-dimensional vector of partial derivatives $\partial \boldsymbol{\phi}_{\mathcal{T},i} / \partial \boldsymbol{\theta}_j$,

- $\partial r_{\boldsymbol{\phi}_{\mathcal{T}}} / \partial \boldsymbol{\phi}_{\mathcal{T}}$ denotes the $k \times |\mathcal{S}||\mathcal{A}|$-dimensional matrix of partial derivatives $\partial r_{\boldsymbol{\phi}_{\mathcal{T},i}} / \partial \boldsymbol{\phi}_{\mathcal{T},j}$,

- and, $\partial \mathcal{L}_{\mathcal{T}}^{\text{test}} / \partial r_{\boldsymbol{\phi}_{\mathcal{T}}}$ denotes the $k$-dimensional gradient vector of $\mathcal{L}_{\mathcal{T}}^{\text{test}}$ with respect to $r_{\boldsymbol{\phi}_{\mathcal{T}}}$.

We will now focus on the term inside of the parentheses in Eq. 12, which is a $k \times k$-dimensional matrix of partial derivatives.

$$
\begin{aligned}
\frac{\partial}{\partial \boldsymbol{\theta}} \left( \frac{\partial r_{\boldsymbol{\theta}}}{\partial \boldsymbol{\theta}} (\mathbb{E}_{\tau}[\boldsymbol{\mu}_{\tau}] - \boldsymbol{\mu}_{\mathcal{D}_{\mathcal{T}}}) \right) &= \sum_{i=1}^{|\mathcal{S}||\mathcal{A}|} \frac{\partial^2 r_{\boldsymbol{\theta}}}{\partial \boldsymbol{\theta}^2} (\mathbb{E}_{\tau}[\boldsymbol{\mu}_{\tau}] - \boldsymbol{\mu}_{\mathcal{D}_{\mathcal{T}}})_i + \left( \frac{\partial}{\partial \boldsymbol{\theta}} \mathbb{E}_{\tau}[\boldsymbol{\mu}_{\tau}] \right) \left( \frac{\partial r_{\boldsymbol{\theta}}}{\partial \boldsymbol{\theta}} \right)^{\top} \\
&= \sum_{i=1}^{|\mathcal{S}||\mathcal{A}|} \frac{\partial^2 r_{\boldsymbol{\theta}}}{\partial \boldsymbol{\theta}^2} (\mathbb{E}_{\tau}[\boldsymbol{\mu}_{\tau}] - \boldsymbol{\mu}_{\mathcal{D}_{\mathcal{T}}})_i + \left( \frac{\partial r_{\boldsymbol{\theta}}}{\partial \boldsymbol{\theta}} \right) \left( \frac{\partial}{\partial r_{\boldsymbol{\theta}}} \mathbb{E}_{\tau}[\boldsymbol{\mu}_{\tau}] \right) \left( \frac{\partial r_{\boldsymbol{\theta}}}{\partial \boldsymbol{\theta}} \right)^{\top}
\end{aligned}
$$

where between the first and second lines, we apply the chain rule to expand the second term. In this expression, we make use of the following notation:

- $\partial^2 r_{\boldsymbol{\theta}} / \partial \boldsymbol{\theta}^2$ denotes the $k \times |\mathcal{S}||\mathcal{A}|$-dimensional matrix of second-order partial derivatives of the form $\partial^2 r_{\boldsymbol{\theta},i} / \partial \boldsymbol{\theta}_j^2$,

- $(\mathbb{E}_{\tau}[\boldsymbol{\mu}_{\tau}] - \boldsymbol{\mu}_{\mathcal{D}_{\mathcal{T}}})_i$ denotes the $i$th element of the $|\mathcal{S}||\mathcal{A}|$-dimensional vector $(\mathbb{E}_{\tau}[\boldsymbol{\mu}_{\tau}] - \boldsymbol{\mu}_{\mathcal{D}_{\mathcal{T}}})_i$,

- $\partial r_{\boldsymbol{\theta}} / \partial \boldsymbol{\theta}$ denotes the $k \times |\mathcal{S}||\mathcal{A}|$-dimensional matrix of partial derivatives of the form $\partial r_{\boldsymbol{\theta},i} / \partial \boldsymbol{\theta}_j$,

- and, $\frac{\partial}{\partial r_{\boldsymbol{\theta}}} \mathbb{E}_{\tau}[\boldsymbol{\mu}_{\tau}]$ is the $|\mathcal{S}||\mathcal{A}|$-dimensional Jacobian matrix of $\mathbb{E}_{\tau}[\boldsymbol{\mu}_{\tau}]$ with respect to the reward function $r_{\boldsymbol{\theta}}$ (we will examine in more detail exactly what this is below).

When substituted back into Eq. 12, the resulting gradient is equivalent to that in Eq. 8 in Section 4. In order to compute this gradient, however, we must take the gradient of the expectation $\mathbb{E}_{\tau}[\boldsymbol{\mu}_{\tau}]$

with respect to the reward function $r_{\boldsymbol{\theta}}$. This can be done by expanding the expectation as follows

$$
\frac{\partial}{\partial r_{\boldsymbol{\theta}}} \mathbb{E}_{\tau}[\boldsymbol{\mu}_{\tau}] = \frac{\partial}{\partial r_{\boldsymbol{\theta}}} \sum_{\tau} \left( \frac{\exp(\boldsymbol{\mu}_{\tau}^{\top} r_{\boldsymbol{\theta}})}{\sum_{\tau'} \exp(\boldsymbol{\mu}_{\tau'}^{\top} r_{\boldsymbol{\theta}})} \right) \boldsymbol{\mu}_{\tau}
$$

$$
= \sum_{\tau} \left( \left( \frac{\exp(\boldsymbol{\mu}_{\tau}^{\top} r_{\boldsymbol{\theta}})}{\sum_{\tau'} \exp(\boldsymbol{\mu}_{\tau'}^{\top} r_{\boldsymbol{\theta}})} \right) (\boldsymbol{\mu}_{\tau} \boldsymbol{\mu}_{\tau}^{\top}) - \frac{\exp(\boldsymbol{\mu}_{\tau}^{\top} r_{\boldsymbol{\theta}})}{(\sum_{\tau'} \exp(\boldsymbol{\mu}_{\tau'}^{\top} r_{\boldsymbol{\theta}}))^{2}} \sum_{\tau'} (\boldsymbol{\mu}_{\tau'} \boldsymbol{\mu}_{\tau}^{\top}) \exp(\boldsymbol{\mu}_{\tau'}^{\top} r_{\boldsymbol{\theta}}) \right)
$$

$$
= \sum_{\tau} P(\tau \mid r_{\boldsymbol{\theta}})(\boldsymbol{\mu}_{\tau} \boldsymbol{\mu}_{\tau}^{\top}) - \sum_{\tau} P(\tau | r_{\boldsymbol{\theta}}) \sum_{\tau'} P(\tau' \mid r_{\boldsymbol{\theta}})(\boldsymbol{\mu}_{\tau'} \boldsymbol{\mu}_{\tau}^{\top})
$$

$$
= \mathbb{E}_{\tau} \left[ (\boldsymbol{\mu}_{\tau} \boldsymbol{\mu}_{\tau}^{\top}) - \sum_{\tau'} P(\tau' \mid r_{\boldsymbol{\theta}})(\boldsymbol{\mu}_{\tau'} \boldsymbol{\mu}_{\tau}^{\top}) \right]
$$

$$
= \mathbb{E}_{\tau}[\boldsymbol{\mu}_{\tau} \boldsymbol{\mu}_{\tau}^{\top}] - \mathbb{E}_{\tau',\tau}[\boldsymbol{\mu}_{\tau'} \boldsymbol{\mu}_{\tau}^{\top}]
$$

$$
= \mathbb{E}_{\tau}[\boldsymbol{\mu}_{\tau} \boldsymbol{\mu}_{\tau}^{\top}] - \mathbb{E}_{\tau}[\boldsymbol{\mu}_{\tau}](\mathbb{E}_{\tau}[\boldsymbol{\mu}_{\tau}])^{\top}
$$

$$
= \mathrm{Cov}[\boldsymbol{\mu}_{\tau}].
$$

