# OpenReview forum: "Few-Shot Intent Inference via Meta-Inverse Reinforcement Learning"
_ICLR.cc/2019/Conference_

### Official Review · AnonReviewer2 · 2018-11-02
**Nice and novel idea but not tested enough**

**Rating:** 4
**Confidence:** 4

**Review:**

The paper defines a new machine learning problem setup by applying the meta-learning concept to inverse reinforcement learning (IRL). The motivation behind this setup is that expert demonstrations are scarce, yet the reward functions of related tasks are highly correlated. Hence, there is plenty of transferrable knowledge across tasks.

Strengths:
--
 * The proposed setup is indeed novel and very ecologically-valid, meaning meta-learning and IRL are natural counterparts for providing a remedy to an important problem.

 * The paper is well-written, technically sound, and provides a complete and to-the-point literature survey. The positioning of the novelty within literature is also accurate.


Weaknesses:
--

 * The major weakness of the paper is that its hypothesis is not tested exhaustively enough to draw sound conclusions. The paper reports results only on the SpriteWorld data set, which is both synthetic and grid-based. Having acknowledged that the results reported on this single data set are very promising, I do not find this evidence sufficient to buy the proposed hypothesis. After all, IRL is meant mainly for real-world tasks where rewards are not easy to model. A single result on a simplified computer game does not shed much light on where an allegedly state-of-the-art model stands toward such an ambitious goal. I would at least like to see results on some manipulation tasks, say on half-cheetah, ant etc.

 * Combination of MaxEnt IRL and MAML is novel. That said, the paper combines them in the most straightforward way, which does not incur any complications that call for technical solution that can be counted as a contribution to science. Overall, I find the novelty of this work overly incremental and its impact potential very limited.

 * A minor issue regarding clarity. Equation 3 is not readable at all. The parameter \phi and the loss \mathcal{L}_{IRL} have not been introduced.

This paper consists of a mixture of strong and weak aspects as detailed above. While the proposed idea is novel and the first results are very promising, I view this work to be at a too early stage to appear in ICLR proceedings as a full-scale paper. I would like to encourage the authors to submit it to a workshop, strengthen its empirical side and resubmit to a future conference.

---

> ### Author Response · Authors · 2018-11-25
> **Response**
>
> Thanks for your review. We have addressed the clarifications requested by the reviewer and added clarifying comments on the additional domains required by the reviewers.
>
> “The major weakness of the paper is that its hypothesis is not tested exhaustively enough to draw sound conclusions … A single result on a simplified computer game does not shed much light on where an allegedly state-of-the-art model stands toward such an ambitious goal. I would at least like to see results on some manipulation tasks, say on half-cheetah, ant etc.
> ..
> Combination of MaxEnt IRL and MAML is novel. That said, the paper combines them in the most straightforward way”
>
> We appreciate the feedback. However, we believe that this criticism is somewhat unfair: the tasks in our evaluation are comparable in complexity or more complex than prior work in IRL. IRL is a difficult problem, and almost all prior IRL papers employ tabular domains. Our domains have raw pixel observations for the reward, which makes them more complex. We summarize the evaluation domains in prior work here:
> Ratliff et al. ICML '06 consider a discretized driving domain, Ziebart et al., AAAI '08 considered a driver route modelling setting with roads featurized by 22 dimensional vectors, Hadfield-Menell et al. NIPS '16 considered a tabular domain and used a featurized state representation with dimensionality 3 or 10, Hadfield-Menell et al. NIPS '17 considered 4 variants of a tabular domain with a featurized state representation of 50 dimensions, Malik et al. ICML '18 considered a tabular domain on the order of a 5 x 5 grid with 4 object classes and 4 objects.
>
> Regardless of what experiments are presented, it is always the case that more experiments on more complex domains provides stronger evidence for an approach. From this perspective, an approach should provide results comparable to prior work in the literature. It is not uncommon for prior work in the IRL domain to demonstrate the benefits their approach in a single domain as demonstrated above.
>
> In addition, to our knowledge there is no prior work on the problem of meta-IRL. There is also no prior work demonstrating IRL on half-cheetah to the best of our knowledge. The only example on Ant comes from (Fu et al. 2017) but this environment is not suited to generating a large diversity of tasks for meta-learning.
>
> “The parameter \phi and the loss \mathcal{L}_{IRL} have not been introduced.”
>
> That equation was meant to provide the definition of the Max-Ent IRL loss. We have fixed this.

---

### Official Review · AnonReviewer3 · 2018-11-07
**Novel solution to an important problem, but needs further details and experimentation**

**Rating:** 4
**Confidence:** 3

**Review:**

This paper attempts to the solve  data-set coverage issue common with Inverse reinforcement learning based approaches - by introducing a meta-learning framework trained on a smaller number of basic tasks. The primary insight here is that there exists a smaller set of unique tasks, the knowledge from which is transferable to new tasks and using these to learn an initial parametrized reward function improves the coverage for IRL. With experiments on the SpriteWorld synthetic data-set, the authors confirm this hypothesis and demonstrate performance benefits - showcasing better correlation with far fewer  number of demonstrations.

Pros:
+ The solution proposed here in novel - combining meta-learning on tasks to alleviate a key problem with IRL based approaches.
The fact that this is motivated by the human-process of learning, which successfully leverages tranferability of knowledges across a group of basic tasks for any new (unseen) tasks, makes it quite interesting.
+ Unburdens the needs for extensive datasets for IRL based approach to be effective
+ To a large extent, circumvents the need of having to manually engineered features for learning IRL reward functions

Cons:
- Although the current formulation is novel, there is a close resemblance to other similar approaches  - mainly, imitation learning. It would be good if the authors could contrast the differences between the proposed approach and approach based on imitation learning (with similar modifications). Imitation learning is only briefly mentioned in the related work (section-2), it would be helpful to elaborate on this. For instance, with Alg-1 other than the specific metric used in #3 (MaxEntIRLGrad), the rest seems close similar to what would be done with imitation learning?
- One of main contributions is avoiding the need for hand-crafted features for the IRL reward function. However, even with the current approach, the sampling of the meta-learning training and testing tasks seem to be quite critical to the performance of the overall solution and It seems like this would require some degree of hand-tuning/picking. Can the authors comment on this and the sensitivity of the results to section of meta-learning tasks and rapid adaption?
- The results are limited, with experiments using only the synthetic (seemingly quite simple)  SpriteWorld data-set. Given the stated objective of this work to extend IRL to beyond simple  cases, one would expect more results and with larger problems/data-sets.
	- Furthermore, given that this work primarily attempts to improve performance with using meta-learned reward function instead of default initialization - it might make sense to also compare with method such as Finn 2017, Ravi & Larochelle 2016.

Minor questions/issues:
> section1: Images are referred to as high dimensional observation spaces, can this be further clarified?
>  section3: it is not immediately obvious how to arrive at eqn.2. Perhaps additional description would help.
> section4.1 (MandRIL) meta-training: What is the impact/sensitivity of computing the state visitation distribution with either using the average of expert demos  or the true reward? In the reported experiments, what is used and what is the impact on results, if any ?
> section4.2: provides an interesting insight with the concept of locality of the prior and establishes the connection with Bayesian approaches.
> With the results, it seems like that other approaches continue to improve on performance with increasing number of demonstrations (the far right part of the Fig.4, 5) whereas the proposed approach seems to stagnate - has this been experimented further ? does this have implications on the capacity of meta-learning ?
> Overall, given that there are several knobs in the current algorithm, a comprehensive sensitivity study on the relative impact would help provide a more complete picutre

---

> ### Author Response · Authors · 2018-11-25
> **Response**
>
> We thank the reviewer for their comments. We have addressed the clarifications requested by the reviewer and added comments regarding requested comparisons.
>
> “there is a close resemblance to other similar approaches  - mainly, imitation learning.”
>
> Behavior cloning is a simple solution, but only tends to succeed with large amounts of data. This is due to the problem of compounding error due to covariate shift which has been studied in prior work (e.g. Ross AISTATS 2010, Ross 2011). Learning reward functions that reason about outcomes, and prioritize entire trajectories over others, can mitigate this effect by avoiding per time-step fitting. The difference between these two approaches has been further discussed in prior work in the literature (e.g. MacGlashan et al. 2015) and IRL has been shown to be a better solution in settings where lots of data is not available (e.g. Finn et al. 2016, Fu et al. 2017).
>
> “Furthermore, given that this work primarily attempts to improve performance with using meta-learned reward function instead of default initialization - it might make sense to also compare with method such as Finn 2017, Ravi & Larochelle 2016”
>
> Thank you for the suggestion. Neither Finn et al. 2017 nor Ravi & Larochelle 2016 actually tackle the inverse reinforcement learning problem, so these methods are not directly applicable. Our method is an extension of Finn et al. 2017 to the IRL setting. An extension of Ravi & Larochelle is likely also possible, but would itself constitute a novel method. We do compare to an alternative meta-learning approach based on recurrent models, which is somewhat similar to Duan et al. 2017 (see Figure 6 red line), and we find that our approach substantially outperforms this alternative method.
>
> “The results are limited, with experiments using only the synthetic (seemingly quite simple) SpriteWorld data-set. Given the stated objective of this work to extend IRL to beyond simple  cases, one would expect more results and with larger problems/data-sets”
>
> The complexity of the experiments presented in the IRL literature are generally less complex than the RL literatures. Our experiments however are able to scale to high-dimensional image inputs, in contrast to many prior methods in IRL (see below). The complexity of the experiments should be judged relative to the literature, rather than a different problem formulation. In the IRL literature, it is not uncommon to report results in domains which we summarize here: Ratliff et al. ICML '06 consider a discretized driving domain, Ziebart et al., AAAI '08 considered a driver route modelling setting with roads featurized by 22 dimensional vectors, Hadfield-Menell et al. NIPS '16 considered a tabular domain and used a featurized state representation with dimensionality 3 or 10, Hadfield-Menell et al. NIPS '17 considered 4 variants of a tabular domain with a featurized state representation of 50 dimensions, Malik et al. ICML '18 considered a tabular domain on the order of a 5 x 5 grid with 4 object classes and 4 objects. We also emphasize that our domain is more challenging than those in prior work as it requires scaling to high-dimensional observations spaces.
>
> “Images are referred to as high dimensional observation spaces, can this be further clarified?”
>
> The observation spaces considered in recent prior IRL work typically operates on features spaces on the order of 10-50 dimensions (e.g. Hadfield-Menell et al. NIPS’16, Hadfield-Menell el al. NIPS’17) or tabular states (e.g. Malik et al. ICML '18). In contrast, our reward function is defined directed on image observations, which in the case of our 84 x 84=7056 images is much higher dimensional.

---

> > ### Comment · AnonReviewer3 · 2018-12-10
> > **Thank you**
> >
> > Thank to the authors for the provided clarifications. I believe this answers my question on resemblance to other approaches and that this sufficiently different.
> >
> > However my concern regarding claim of "avoiding the need for hand-crafted features for the IRL reward function"  by the suggested approach is still persists. I am of the opinion that this complexity is now shifted elsewhere. I would still suggest a more rigorous exploration of the sensitivity to these parameters, along with a detailed study of impact on solution quality is warranted. In addition including more experimental results providing a comprehensive picture of the proposed solution would help improve the paper.  Given these reasons, I intend to leave with my original score unchanged.

---

### Official Review · AnonReviewer1 · 2018-11-10
**An interesting application of MAML to Inverse RL but lacks rigorousness and persuasive experimental results**

**Rating:** 3
**Confidence:** 5

**Review:**

This paper aims to address the problem of lacking sufficient demonstrations in inverse reinforcement learning (IRL) problems. They propose to take a meta learning approach, in which a set of i.i.d. IRL tasks are provided to the learner and the learner aims to learn a strategy to quickly recover a good reward function for a new task that is assumed to be sampled from the same task distribution. Particularly, they adopt the gradient-based meta learning algorithm, MAML, and the maximal entropy (MaxEnt) IRL framework, and derive the required meta gradient expression for parameter update. The proposed algorithm is evaluated on a synthetic grid-world problem, SpriteWorld. The experimental results suggest the proposed algorithm can learn to mimic the optimal policy under the true reward function that is unknown to the learner.

Strengths:

1) The use of meta learning to improve sample efficiency of IRL is a good idea.
2) The combination of MAML and MaxEnt IRL is new to my knowledge.
3) Providing the gradient expression is useful, which is the main technical contribution of this paper. (But it needs to be corrected; see below.)
4) The paper is well motivated and clearly written "in a high level" (see below).

Weakness:

1) The derivation of (5) assumes the problem is tabular, and the State-Visitations-Policy procedure assumes the dynamics/transition of the MDP is known. These two assumption are rather strong and therefore should be made explicitly in the problem definition in Section 3.

2)  Equation (8) is WRONG. The direction of the derivation takes is correct, but the final expression is incorrect. This is mostly because of the careless use of notation in derivation on p 15 in the appendix (the last equation), in which the subscript i is missed for the second term. The correct expression of (8) should have a rightmost term in the form  (\partial_\theta r_\theta) D  (\partial_\theta r_\theta)^T, where D is a diagonal matrix that contains \partial_{r_i} (\E_{\tau} [ \mu_\tau])_i and i is in 1,...,|S||A|.

3) Comparison with imitation learning and missing details of the experiments.
a) The paper assumes the expert is produced by the MaxEnt model. In the experiments, it is unclear whether this is true or not, as the information about the demonstration and the true reward is not provided.
b) While the experimental results suggest the algorithm can recover the similar performance to the optimal policy of the true reward function, whether this observation can generalize outside the current synthetic environment is unclear to me. In imitation learning, it is known that the expert policy is often sub-optimal, and therefore the goal in imitation learning is mostly only to achieve expert-level performance. Given this, the way this paper evaluate the performance is misleading and improper to me, which leads to an overstatement of the benefits of the algorithm.
c) It would be interesting to compare the current approach with, e.g., the policy-based supervised learning approach to imitation learning (i.e. behavior cloning).

4) The rigorousness in technicality needs to be improved. While the paper is well structured, the writing at the mathematical level is careless, which leads to ambiguities and mistakes (though one might be able to work out the right formula after going through the details of the entire paper). Below I list a few points.
    a) The meta-training set {T_i; i=1,...,N} and the meta-test set {T_j; i=1,...,M} seems to overload the notation. I suppose this is unintentional but it may appear that the two sets share the first T_1,.., T_M tasks, e.g., when N>=M, instead of being disjoint.
    b) The set over which the summation is performed in (4) is unclear; alpha in (4) is not defined, though I guess it's a positive step size.
    c) On p4, "we can view this problem as aiming to learn a prior over the intentions of human demonstrators" is an overstatement to me. At best, this algorithm learns a prior over rewards for solving maximal entropy IRL, not intention. And the experiment results do not corroborate  the statement about "human" intention.
    d) On p4,  "since the space of relevant reward functions is much smaller than the space of all possible rewards deﬁnable on the raw observations" needs to be justified. This may not be true in general, e.g., learning the set of relevant functions may require a larger space than learning the reward functions.
    e) The authors call \mu_\tau the "state" visitation, but this is rather confusing, as it is the visiting frequency of state and action (which is only made clear late in the appendix).
    f) On p5, it writes "... taking a small number of gradient steps on a few demonstrations from given task leads" But the proposed algorithm actually only takes "one" gradient step in training.
    g) The convention of derivatives used in the appendix is the transpose of the one used in the main paper.

Minor points:
1) typo in (2)
2) p_\phi is not defined, L_{IRL} is not defined, though the definition of both can be guessed.
3) T^{tr} seems to be typo in (11)
4) A short derivation of (2) in the Appendix would be helpful.

---

> ### Author Response · Authors · 2018-11-25
> **Response**
>
> We thank the reviewer for their comprehensive review. We have addressed the clarification required by the reviewer. We have also requested some important clarifications on certain comments made by the reviewer.
>
> “The paper assumes the expert is produced by the MaxEnt model.”
>
> We have made the MaxEnt modeling assumption more explicit in the paper (see page 3). In the IRL literature, the MaxEnt model is a standard assumption (Ziebart et al. 2008, Levine et al. 2012, Huang et al. 2014, Ho et al. 2016, Finn et al. 2016, Fu et al. 2017) as it allows for sub-optimal demonstrations and has a connection to maximum likelihood estimation.
>
> “In imitation learning, it is known that the expert policy is often sub-optimal, and therefore the goal in imitation learning is mostly only to achieve expert-level performance. Given this, the way this paper evaluate the performance is misleading and improper to me, which leads to an overstatement of the benefits of the algorithm.”
>
> Can you provide details on how you believe the evaluation should be done? Using value difference to evaluate the quality of the rewards follows prior work in inverse reinforcement learning (Levine et al. NIPS '11, Wulfmeier et al, 2016, Brown et al. AAAI 2018), and therefore seemed like the most appropriate evaluation metric. Value difference measures the difference between the learned policy’s performance and the expert, which seems to be a good measure of whether the policy achieves expert-level performance. We would be happy to add some other metric that you might recommend, if you have specific metrics in mind.
>
> “Equation (8)...“
>
> We have fixed this. You are correct that there is a missing index over i that was dropped by mistake in the appendix. Thank you for pointing that out.
>
> “a) The meta-training set {T_i; i=1,...,N} and the meta-test set {T_j; i=1,...,M} seems to overload the notation. I suppose this is unintentional but it may appear that the two sets share the first T_1,.., T_M tasks, e.g., when N>=M, instead of being disjoint.”
>
> We have addressed this in the paper by stating that meta-train and meta-test sets are explicitly disjoint.
>
> “b) The set over which the summation is performed in (4) is unclear; alpha in (4) is not defined, though I guess it's a positive step size.”
>
> We have clarified this to indicate that alpha is a step size.
>
> “c) On p4, "we can view this problem as aiming to learn a prior over the intentions of human demonstrators" is an overstatement to me. At best, this algorithm learns a prior over rewards for solving maximal entropy IRL, not intention. And the experiment results do not corroborate  the statement about "human" intention.”
>
> We provided this sentence to give some intuition for our approach (the first word in the quoted sentence is “Intuitively”). We clarified to this sentence to use the word “reward” in place of “intentions” and “expert” instead of “human”.
>
> “d) On p4,  "since the space of relevant reward functions is much smaller than the space of all possible rewards deﬁnable on the raw observations" needs to be justified. This may not be true in general, e.g., learning the set of relevant functions may require a larger space than learning the reward functions.”
>
> We clarified this sentence in the paper. We are referring to reward functions that can explain a particular behavior. In this sense, it is a strict subset of the reward functions we can define.
>
> “e) The authors call \mu_\tau the "state" visitation, but this is rather confusing, as it is the visiting frequency of state and action (which is only made clear late in the appendix).”
>
> This terminology comes from the original MaxEnt IRL paper (see Ziebart 2008). We agree however, and have clarified this in the paper.
>
> “f) On p5, it writes "... taking a small number of gradient steps on a few demonstrations from given task leads" But the proposed algorithm actually only takes "one" gradient step in training.”
>
> To clarify, in our experiment, we use one gradients step during meta-training, but up to 20 at meta-test. We discuss this in the paper. We note that is it is possible to take more than one gradient step at meta-training time although it is computationally more expensive.
>
> “g) The convention of derivatives used in the appendix is the transpose of the one used in the main paper.”
>
> We will correct this.
>
> “3) T^{tr} seems to be typo in (11)”
>
> To clarify, this is not a typo. It is consistent with our notation in the preliminary section. In meta-learning, there is a meta-training and meta-testing dataset which consists of tasks. For each task, there is T^{tr} and T^{test}, which are training and test points. It is easy to see that few-shot learning is one such example of this.

---

> > ### Author Response · Authors · 2018-12-10
> > **Comment: the value difference metric is "normalized"**
> >
> > We would just like to point out that the value difference metric is "normalized". When the value difference is zero, the recovered policy is optimal. The difference is computed with respect to V*, which we explain in the paper. There seems to have a been a persistent misunderstanding of this metric, which we hope is now clarified.

---

> > > ### Comment · AnonReviewer1 · 2018-12-10
> > > **RE: Comment: the value difference metric is "normalized"**
> > >
> > > I think my point is that in imitation learning with suboptimal experts (which is usually the case), a normalized metric should be relative to the performance of the suboptimal expert, not the optimal policy for the original RL. Being normalized wrt the true optimal policy does not provide any calibration, because the suboptimal expert that provides the demonstrations can also have a large value difference wrt V*.

---

> > > > ### Author Response · Authors · 2018-12-10
> > > > **Thanks for your continued feedback, but imitation learning is not the same as inverse RL**
> > > >
> > > > Our paper is on inverse reinforcement learning. The goal of this setting to learn the cost function of another agent through observing behavior. Evaluation of the learned cost function should naturally be with respect to the original cost function. Since we are trying to explicitly recover the cost function of the expert, the value difference metric seems to us reasonable. We emphasize again that this comes from prior work.
> > > >
> > > > This problem statement is different from the imitation setup. In fact, imitation learning very often does not recover a cost function at all.
> > > >
> > > > We would like to address your concern and thank you for your time, but can you provide a more specific description of what normalized metric are you suggesting?

---

### Meta-Review · Area_Chair1 · 2018-12-15
**Well-motivated idea but execution and analysis is not convincing**

**Confidence:** 5
**Recommendation:** Reject

**Metareview:**

This work proposes to use the MAML meta-learning approach in order to tackle the typical problem of insufficient demonstrations in IRL.

All reviewers found this work to contain a novel and well-motivated idea and the manuscript to be well-written. The combination of MAML and MaxEnt IRL is straightforward, as R2 points out, however the AC does not consider this to be a flaw given that the main novelty here is the high-level idea rather than the technical details.

However, all reviewers agree that for this paper to meet the ICLR standards, there has to be an increase in rigorousness through (a) a more close examination of assumptions, sensitivity of parameters and connections to imitation learning (b) expanding the experimental section.